



# Winter season Southern Ocean distributions of climate-relevant trace gases

Li Zhou[1], Dennis Booge[1], Miming Zhang[2], Christa A. Marandino[1]

[1]Research Division 2-Biogeochemistry, GEOMAR Helmholtz Centre for Ocean Research Kiel, Kiel, Germany
[2]Key Laboratory of Global Change and Marine-Atmospheric Chemistry of Ministry of Natural Resources (MNR), Third Institute of Oceanography, Xiamen, PR China.

*Correspondence to*: Li Zhou (lzhou@geomar.de)

**Abstract.** Climate-relevant trace gas air-sea exchange exerts an important control on air quality and climate, especially in remote regions of the planet such as the Southern Ocean. It is clear that polar regions exhibit seasonal trends in
productivity and biogeochemical cycling, but almost all of the measurements there are skewed to summer months. If we want to understand how the Southern Ocean effects the balance of climate through trace gas air-sea exchange, it is essential to expand our measurement database over greater temporal and spatial scales, including all seasons. Therefore, in this study, we report measured concentrations of dimethylsulphide (DMS, and related sulphur compounds) and isoprene in the Atlantic sector of the Southern Ocean during the winter to understand the spatial and
temporal distribution in comparison to current knowledge and climatological calculations for the Southern Ocean. The observations of isoprene are the first in the winter season in the Southern Ocean. We find that concentrations and fluxes of DMS and isoprene in the investigated area are generally lower than those presented or calculated in currently used climatologies and models. More data is urgently needed to better interpolate climatological values and validate process-oriented models, as well as to explore how finer measurement resolution, both spatially and temporally, can
influence air-sea flux calculations.

## 1 Introduction

Despite the low abundance of trace gases in the atmosphere, their strong chemical reactivity and interactions with radiation have an important influence on air quality and the climate system (Monson and Holland, 2001). For example,
a wide variety of trace gases, such as carbon dioxide ($CO_2$), methane, and nitrous oxide, trap heat and contribute to global atmospheric warming (Liss, 2007). The ocean plays an important role in regulating the sources and sinks of trace gases and, thus, strongly impacts the biogeochemical cycles and budget of reactive trace gases in the global atmosphere (Houghton et al., 2001; Liss et al., 2014; Vallina and Simo, 2007). Studying the air-sea exchange of climate-relevant trace gases can improve the understanding of their effect on climate (Emerson et al., 1999; Liss et al.,
2014). Here we focus on two typical marine biogenic gases, i.e. dimethylsulphide (DMS) and isoprene, which have a significant influence on aerosols and climate in remote areas of the world (Carpenter et al., 2012; Lovelock et al., 1972).



DMS was hypothesized to influence climate by regulating aerosols and clouds, thus, decreasing the amount of solar

radiation reaching the Earth's surface, known as the CLAW hypothesis (Charlson et al., 1987). DMS is produced from the degradation of dimethyl sulfoniopropionate (DMSP), which is formed in the cells of marine organisms (Cantoni and Anderson, 1956; Curson et al., 2011). DMSP producers include phytoplankton (e.g. coccolithophores, dinoflagellates, diatoms), angiosperms, macroalgae, and some corals (Broadbent et al., 2002; Keller et al., 1989; Otte et al., 2004; Van Alstyne, 2008; Yoch, 2002). DMSP is cleaved to DMS by bacteria and phytoplankton (Curson et al.,

2011; Stefels et al., 2007). DMS produced in the surface ocean can be consumed in the ocean, be oxidized to form dimethylsulphoxide (DMSO), or be released to the atmosphere (Vogt and Liss, 2009). Only about 10% of the DMS produced in the surface ocean is released into the atmosphere (Archer et al., 2001). DMS in the atmosphere is oxidized to form sulfuric acid and methanesulphonic acid (McArdle et al., 1998) by hydroxyl radicals (OH; 66 %), nitrate ($NO_3$; 16 %) and bromine monoxide radicals (BrO; 12 %) globally (Chen et al., 2018), with an atmospheric lifetime of

approximately one day (Kloster et al., 2006). These DMS by-products can form aerosols (new particles) (Kulmala et al., 2000) or lead to growth of existing aerosol particles (Andreae and Crutzen, 1997; von Glasow and Crutzen, 2004), aiding the formation of cloud condensation nuclei (CCN) (Charlson et al., 1987; Sanchez et al., 2018). Especially in the remote marine boundary layer (MBL) of the North Atlantic and Polar Oceans (e.g. the Southern Ocean (SO)), DMS-derived  non-sea salt sulphate particles account for 33% and 7%-65% (7%-20% in winter and 43%-65% in

summer), respectively (Jackson et al., 2020; Korhonen et al., 2008; Sanchez et al., 2018). Mahmood et al. (2019) show that the mean cloud radiative forcing in the Arctic could increase between 108 % and 145 % from 2000 to 2050 because of increasing Arctic DMS emissions. The global radiative effect of DMS is calculated to be -1.69 to -2.03 W m$^{-2}$ at the top of the atmosphere (Fiddes et al., 2018; Mahajan et al., 2015; Thomas et al., 2010).

Isoprene is the most important biogenic volatile organic compound (BVOCs) in the atmosphere, accounting for 50% of all BVOCs coming from terrestrial ecosystems (Guenther et al., 2012; Laothawornkitkul et al., 2009; Sharkey et al., 2008). It impacts the climate system and oxidant chemistry in the atmosphere via secondary organic aerosol (SOA) formation and interaction with OH and the ozone cycle (Claeys et al., 2004; Guenther et al., 1995; Went, 1960). Most isoprene in the atmosphere is produced by terrestrial ecosystems, but isoprene is also produced in the ocean (Bonsang

et al., 1992). All species of phytoplankton and seaweed (Bonsang et al., 1992; Shaw et al., 2003), and some species of marine bacteria (Exton et al., 2013), produce isoprene. The global marine flux of isoprene is reported to range from 0 to 11 Tg C yr$^{-1}$ (Booge et al., 2016), with more extreme values reported by Tran et al. (2013) and Kameyama et al. (2014) of 0.51- 16.53 Tg C yr$^{-1}$ in June–July 2010 in the Arctic and 9.05- 34.96 Tg C yr$^{-1}$ in the productive Southern Ocean during austral summer 2010/2011, respectively. Despite that these values are significantly less than the

terrestrial flux (400-600 Tg C yr$^{-1}$; (Arneth et al., 2008; Arnold et al., 2009; Baker et al., 2000; Guenther et al., 2006), emitted isoprene in the marine atmosphere plays an important role in the chemistry there, as it is extremely short-lived (lifetime of ~ 1 hour due to reaction with OH radicals). Terrestrial isoprene is unlikely to reach the marine boundary layer and all the marine isoprene emitted will quickly react (Booge et al., 2018; Palmer and Shaw, 2005), influencing local climate and air quality (Claeys et al., 2004).




The fluxes of marine derived trace gases are an important parameter in atmospheric budgets and, subsequently, for the evaluation of their climate implications. Typically, ocean-atmosphere fluxes are calculated by multiplying the wind speed-based gas transfer velocity by the bulk phases of the air/sea concentration difference, as follows: $F = k\Delta C$ (Liss and Slater, 1974, see methods section below). Therefore, trace gas fluxes are computed using measured wind

speed, measured atmospheric concentrations, and measured seawater concentrations. Often, only seawater concentrations are used and the atmospheric values are either set to zero or to a constant level, but this can lead to large uncertainties in calculated fluxes (Lennartz et al., 2017; Zhang et al., 2020). Thus, having accurate, repeated measurements of trace gases in the surface ocean, as well as in the marine boundary layer, over a range of spatial and temporal scales is necessary for high quality flux computations.

The Global Surface Seawater DMS database contains 89,324 measurements of surface ocean DMS concentration from 1972 to 2019 (https://saga.pmel.noaa.gov/dms/). Concentrations of oceanic DMS within the database range between 0 and 295 nmolL$^{-1}$. The broader Southern Ocean (latitude range -35°N to -75°N) is represented by 21,580 points for all seasons, but only 158 points (0.7%) are from the austral winter. All three DMS climatologies draw heavily on this database (Hulswar et al., 2021; Kettle et al., 1999; Lana et al., 2011). This DMS data product and resulting

climatologies, along with those for other trace gases (MEMENTO, SOCAT, HalOcAt, etc), are extremely important for model input and validation. If the data products contain the appropriate data, they can resolve sea water concentrations spatially and temporally (e.g. seasonally), as well as begin to point to interannual variability and trends. Therefore, these valuable assets must be equipped with as much data from all regions and seasons as possible. Lana et al. (2011), the most currently used DMS climatology, shows that DMS concentrations typically range from 1-7

nmol L$^{-1}$, with higher concentrations occurring in the high latitude regions with strong seasonality. The highest DMS concentrations appear in the high latitude provinces of the North Atlantic and North Pacific in summer, with DMS concentrations generally increasing with temperature and light and sometimes exceeding 20 nmol L$^{-1}$ (Lana et al., 2011). In the temperate and subtropical provinces, the seasonality becomes weaker, until around the equator, where there is no obvious seasonal change. The transition to the southern subtropical zone shows weaker seasonal changes,

but in austral summer, the Southern Ocean circumpolar regions display a hotspot of DMS concentrations (>10 nmol L$^{-1}$) (McTaggart and Burton, 1992). Lana et al. (2011) estimated that approximately 28.1 Tg S are transferred from the oceans into the atmosphere annually in the form of DMS. The natural sulphur emission has been estimated as 38-89 Tg of sulphur yr$^{-1}$ (Andreae, 1990), of which marine DMS emission contributes 30-70%. Although there were many field campaigns performed, the obtained oceanic DMS data is still insufficient, leaving uncertainties about sea-to-air

DMS fluxes, especially during the winter season. In the Lana climatology, Southern Ocean data is skewed to spring and summer and is spatially non-uniform, requiring the use of interpolation/extrapolation techniques. Thus, it is unavoidable that large discrepancies between fluxes calculated in situ versus those in the climatology are found (even at levels as high as 47–76% (Zhang et al., 2020)). Better spatial and temporal coverage of in situ measurements are needed for adequate computations of the influence of DMS on global climate.




Marine production and emission of isoprene were first described by Bonsang et al. (1992). Currently published isoprene seawater values from the world oceans generally range from below 1 to 200 pmol L$^{-1}$ (Baker et al., 2000; Bonsang et al., 1992; Booge et al., 2016; Broadgate et al., 1997; Broadgate et al., 2004; Hackenberg et al., 2017; Li et al., 2019; Matsunaga et al., 2002; Milne et al., 1995; Ooki et al., 2015; Zindler et al., 2014). The highest reported

concentration of isoprene, 541 pmol L$^{-1}$, in the surface ocean was found in the Arctic Ocean in June-July 2010 (Tran et al., 2013). Marine isoprene concentrations in the eastern North Pacific range from 2 to 6.5 pmol L$^{-1}$, which is at the lower end compared to the world oceans (Moore and Wang, 2006). Additionally, concentrations of marine isoprene show strong seasonal changes in regions with strong seasonal variations in phytoplankton abundances, as e.g. in the East China Sea (Li et al., 2018). Booge et al. (2016) improved the predictive capability of the earlier model from

Palmer and Shaw (2005) by using phytoplankton functional type dependent isoprene production rates, but the model is still limited as it cannot resolve changes in isoprene emissions on short time scales of hours or days. Additionally, the model is validated with a very sparse dataset presently, which cannot resolve seasonal changes in isoprene concentrations for the world oceans. Especially in the Southern Ocean, which is thought to be a hot spot of trace gas emissions during austral summer, data is limited. There are no observations during winter published for this area.

Therefore, it is of fundamental importance to increase the dataset of marine isoprene concentrations to understand the magnitude of the influence of marine isoprene emissions on atmospheric processes over the Southern Ocean.

The Southern Ocean is a typical high-nutrient and low-chlorophyll area due to iron limitation, which exerts a strong influence on global biogeochemical cycles and air-sea gas fluxes (Hauck et al., 2013; Zhang et al., 2017). Knowledge

of the general biological productivity and circulation patterns of the area has made great advances; however, it is still difficult to resolve the small-scale dynamics of gases in the surface (Tortell and Long, 2009), especially during the wintertime. If we want to understand how the Southern Ocean effects on the balance of climate through trace gas air-sea exchange, it is essential to expand our measurement database over greater temporal and spatial scales, including all seasons. Therefore, in this study we measured the concentrations of DMS, its precursor DMSP and oxidation

product DMSO, and isoprene in the Southern Ocean during the austral winter season to gain information on the spatial and temporal distribution in comparison to current knowledge and climatological calculations for the Southern Ocean.

## 2 Methods

### 2.1 Cruise Description

The measurements were performed on the Southern oCean seAsonaL Experiment (SCALE) cruise aboard the *S.A.*

*Agulhas ll*. The cruise started from Cape Town, Republic of South Africa (RSA) on 18 July 2019 (DOY 199), crossed the Southern Ocean to the ice edge and returned from the ice area on 28 July 2019 (DOY 209) to dock in Cape Town on 11 August 2019 (DOY 223) via the East London port from 7-10 August 2019 (DOY 219 - 222) (33ºS~58ºS/2ºW~26ºE, Fig 1). Wind speeds ranged between 1.2 and 29.4 m s$^{-1}$ over the cruise. Air mass back trajectories show that the air was of oceanic origin for most of the cruise. Air temperatures ranged from -19.5 to +18

ºC, sea surface temperatures (SSTs) were from -1.8 to +20.4 ºC, and salinity was 19.4-35.3 over the cruise track.

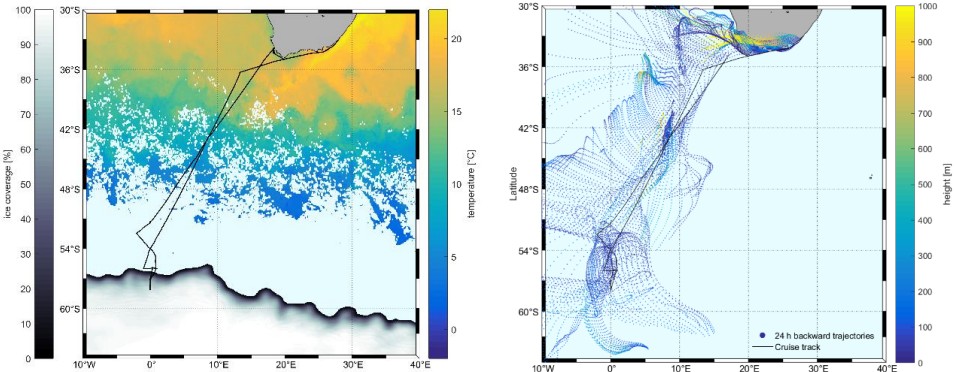

**Figure 1: The cruise track (black) superimposed on satellite data. Left panel – Sea surface temperature (SST) and ice coverage (%). SST and sea ice data are derived from satellite (12 July to 12 August 2019; SST - Naval Oceanographic Office, 2008; Sea ice data - UK Met Office, 2012.). Right panel – 24h air mass back trajectories starting at 50 m height from HYbrid Single Particle Lagrangian Integrated Trajectory (HYSPLIT) using the meteorological fields from the National Centers for Environmental Prediction - Global Data Assimilation System (NCEP-GDAS). The colour shows the average height of the trajectory.**

### 2.2 Sampling

Discrete surface seawater samples for DMS and isoprene were taken bubble-free using transparent 60 mL glass vials (Chromatographie Handel Müller, Fridolfing, Germany) from the underway pump supply. The water samples were kept in a dark, insulated box and analysed within 2 hours. After analysis, DMSP was converted into DMS using sodium hydroxide (NaOH) pellets (≥99%, Carl Roth™ GmbH, Karlsruhe, Germany) and stored for DMSP and DMSO measurements back in the on-shore lab.

Our sampling frequency for DMS and isoprene was one sample per hour at the beginning of the cruise. We changed to one sample per 30 minutes during the last two days of the cruise, because the concentrations of DMS and isoprene were more variable in the coastal area. DMSP/DMSO samples were taken every 2 hours. We obtained 384 discrete samples of DMS and isoprene, and 204 samples of DMSP/DMSO during the cruise. During the cruise, the seawater pumping system was stopped due to the presence of sea ice on 27-28 July 2019 (DOY 208 - 209) and while in port of East London (7-10 August 2019, DOY 219 - 222), resulting in periods with missing data.

We also performed continuous shipboard underway measurements of surface water and lower atmospheric DMS using a home-made purge and trap sampler coupled with a time of flight mass spectrometer system (TOF-MS 3000, Guangzhou Hexin Instrument Co., Ltd., China) (Zhang et al., 2019). Seawater and air samples were introduced continuously to the system through the ship's seawater pump system and air sampler inlet located at the bow at approximately 18 m above the sea surface. A black antistatic tube (1/4'' O. D., 95m) was used to transport the air sample to the laboratory. Every 10 min, we obtained a pair of DMS data points (one in seawater and one in the atmosphere). For seawater sample measurements, we purged a 5 mL aliquot with 65 mL min$^{-1}$ of high purity nitrogen



(N$_2$) for 5.5 min. For atmospheric DMS measurements, the air sample was trapped under the mean flow at 65 mL min$^{-1}$ for 3.5 min. The concentrated air sample was injected to the TOF-MS, then 2 mins later the concentrated water sample was injected to TOF-MS. The atmospheric and seawater DMS limit of detections (LOD) were 32 pptv and 0.07 nmol L$^{-1}$, respectively.

### 2.3 Analysis

DMS/P/O and isoprene were analysed by gas chromatography-mass spectrometry (GC-MS) coupled to a purge and trap system. Headspace within each sample was made by injecting 10 mL of helium into the vial. Isoprene was fully removed from the remaining 50 mL sample (>99%) with helium at a flow rate of 70 mL min$^{-1}$ for 15 min at room temperature (RT). Purge efficiency for DMS was less than 100% and dependent on the seawater temperature, but the data was corrected for this effect (Figure S1). Gaseous deuterated isoprene (isoprene-d5; 98%) was used as internal standard and injected through a 500 µL Sulfinert® stainless steel sample loop (1/16'' O.D., Restek, Bad Homburg, Germany). The sample flow was dried using a Nafion® membrane dryer (counter flow: N$_2$, 180 mL min$^{-1}$, Perma Pure, Ansyco GmbH, Karlsruhe, Germany). After purging, DMS and isoprene were trapped in a Sulfinert® stainless steel trap cooled with liquid N$_2$. The sample was injected into the GC by immersion in hot water. Retention times for DMS and isoprene (m/z: 61, 62; 67, 68) were 5.0 and 5.3 min. For analysis of DMSP, 10 mL of the DMS sample was transferred to brown glass vials (Chromatographie Handel Müller, Fridolfing, Germany). After the DMSP analysis, DMSO was converted into DMS by adding cobalt dosed sodium borohydride (NaBH$_4$) (90%, Sigma-Aldrich Chemie GmbH, Taufkirchen, Germany) and analysed immediately with the same technique as mentioned above. Liquid standards and an internal standard were used to calibrate the system for DMS and isoprene every day during the analysis on board. Liquid calibrations were performed every measuring day for the DMSP/DMSO analysis in the lab. The given LODs of this system are 10 times the standard deviation of the baseline noise, which are 1.8×10$^{-13}$ mol and 5.5×10$^{-13}$ mol for DMS and isoprene, respectively.

Continuous underway measurements of SST and salinity, as well as wind speed and direction, air temperature, pressure, and global radiation were recorded from the ship's pumped seawater supply and the meteorological tower, respectively.

### 2.4 Calculation of air-sea flux

The air-sea fluxes of all gases were calculated with Eq. (1):

$$F = (1-A)k \cdot \Delta C = (1-A)k \cdot \left(Cw - \frac{Ca}{H}\right) \tag{1}$$

where $F$ is the flux (mass area$^{-1}$ time$^{-1}$), $A$ is fraction of sea surface covered by ice, $\Delta C$ is the concentration difference between air ($Ca$) and water ($Cw$), $k$ is the gas exchange coefficient (m s$^{-1}$) in water (Liss and Slater, 1974), and $H$ is the Henry's law coefficient used to calculate gas solubility. The gas exchange coefficient for the gases of interest is usually approximated as the water-air side transfer velocity, $k_w$. We use the following parametrizations derived from dual tracer (Nightingale et al., 2000, N00) and eddy covariance direct measurement of air-sea DMS transfer (Zavarsky et al., 2018, Z18) to calculate the DMS gas transfer velocity following Eq. (2) and Eq. (3)





$$k_{DMS,N00} = (0.222 \cdot U^2 + 0.333 \cdot U)\left(\frac{Sc}{660}\right)^{-0.5} \tag{2}$$

$$k_{DMS,Z18} = (2.00 \cdot U + 0.94)\left(\frac{Sc}{660}\right)^{-0.5} \tag{3}$$

We use the Wanninkhof (1992, W92) and Wanninkhof (2014, W14) formulations, based on a synthesis of tracer, wind-wave tank, radon, and radiocarbon studies, to determine $k_{isoprene}$ (Eqs. 4 and 5),

$$k_{isoprene,W92} = 0.31 \cdot U^2 \left(\frac{Sc}{660}\right)^{-0.5} \tag{4}$$

$$k_{isoprene,W14} = 0.251 \cdot U^2 \left(\frac{Sc}{660}\right)^{-0.5} \tag{5}$$

where $U$ is the wind speed at 10 m height and $Sc$ is the Schmidt number. The wind was measured at 18 m height and
converted to 10 m using,

$$\frac{U_x}{U_{10}} = \left(\frac{Z_x}{Z_{10}}\right)^P \tag{6}$$

where $U_x$ is the observed wind speed at 18 m, $Z_x$ and $Z_{10}$ are heights of 18 and 10 m, respectively, and $P$ depends on atmospheric stability and underlying surface characteristics and is set to 0.11 (Hsu et al., 1994). $Sc$ is defined as the ratio of the kinematic viscosity of water to the diffusion coefficient of gas in water and 660 represents $CO_2$ in seawater
at 20°C. We estimate $Sc$ of DMS and isoprene following Wanninkhof (2014) and Palmer and Shaw (2005), respectively:

$$Sc_{DMS} = 2855.7 - 177.63\,Tc + 6.0438\,Tc^2 - 0.11645\,Tc^3 + 0.00094743\,Tc^4 \tag{7}$$

$$Sc_{isoprene} = 3913.15 - 162.13\,Tc + 2.67\,Tc^2 - 0.012\,Tc^3 \tag{8}$$

where $Tc$ is SST (°C).

In addition, for DMS, the partitioning of the gas transfer coefficient between air-side and water-side control ($\gamma_a$) can change due to low SSTs and from moderate wind speeds (McGillis et al., 2000). Thus, we consider both water-side and air-side control when calculating DMS fluxes,

$$F = (1 - A)\,k_w(1 - \gamma_a) \cdot \Delta C \tag{9}$$

where A is the fraction of sea ice cover, and $\gamma_a$ is calculated as,

$$\gamma_a = \frac{1}{1 + \frac{k_a}{\alpha k_w}} \tag{10}$$

where $k_a$ is air-water side transfer coefficient and $\alpha$ is the Ostwald solubility coefficient. These parameters were calculated as described in McGillis et al. (2000) and the references therein,

$$k_a \approx 659 U_{10}\left(\frac{M}{M_{H_2O}}\right)^{-0.5} \tag{11}$$

$$\alpha = e^{\left[\frac{3525}{T(K)} - 9.464\right]} \tag{12}$$

where $M$ is the molecular weight of DMS or $H_2O$ and $T$ is the sea water temperature ($K$). Finally, as no atmospheric measurements of isoprene were obtained, we assume $Ca$ of isoprene is zero in the remote MBL due to its very short lifetime. Two wind speed-based gas transfer parameterizations for each gas were used and the respective fluxes compared to each other and to existing climatologies or model calculations. For DMS, N00 was used for direct comparison to the Lana climatology and Z18 because it is known that DMS exhibits mostly interfacial gas transfer,
which is more accurately described with a linear wind speed dependence. For isoprene, we used parameterizations





with a quadratic dependence on wind speed, since it is less soluble than DMS and likely to have more influence from bubble mediated gas transfer. W92 was chosen for direct comparison with Palmer and Shaw (2005) and Booge et al. (2016), but the more accurate version of this parameterization is W14 and, thus, it was also used for comparison.

### 2.5 Data analysis

The data was tested for normal distribution using the Kolmogorov-Smirnov test and was determined to be non-normally distributed. Outliers were identified as deviating more than three times from the standard deviation of the mean. We used Spearman correlation analysis to identify correlation coefficients between DMS, DMSP and DMSO. The F-statistic, p-value (significance), R (correlation coefficient) and the $R^2$ (variance proportionality) were calculated to test for a linear correlation between two variables. All statistical analyses were performed using Matlab

(https://www.mathworks.com/).

### 3 Results and Discussion

### 3.1 Comparison of DMS measurements by GC-MS and TOF-MS

Simultaneous measurements of surface seawater DMS concentrations during SCALE were performed using both the GC-MS and TOF-MS. We ensured method comparability by using the same DMS standards (both gas and liquid)

onboard. We collected 361 GC-MS samples and 2245 TOF-MS samples during the *in-situ* observations. Although the detection limit of the GC-MS is lower than that of the TOF-MS, the time resolution of the TOF-MS is higher. Considering the different time resolution of the two instruments, we only compare the data of samples taken at the same time (258 data points). The datasets were found to be correlated and agreed well with each other (Figure 2a, p< 0.01, slope = 0.91, R= 0.67). The median and mean values of TOF-MS observations are higher, 42.5% and 12.9%,

respectively, than those by GC-MS (Figure 2b). This may be due to the fact that the TOF-MS does not have a column that separates isomers and cannot distinguish between compounds of the same mass number, making the mean and median values higher than GC-MS. Alternatively, we observed that 95% of the DMS concentrations in both instruments were less than 1.7 nmol $L^{-1}$ and that in this concentration range, the slope of the fit is slightly lower than the 1:1 line. This may reflect added uncertainty from the GC-MS purge efficiency correction. The regions of the

cruise track corresponding to these lower concentrations were the coldest regions encountered, corresponding to the highest solubilities. Thus, the GC-MS values may be too low in this concentration range. As it is not clear which set of measurements were incorrect and given the very good agreement, we decided to use the data in following way: Fluxes were computed over the entire cruise track using GC-MS measurements for seawater DMS concentrations as the same instrument was used to perform DMSP and DMSP measurements. TOF-MS measurements were used for

atmospheric mixing ratios.

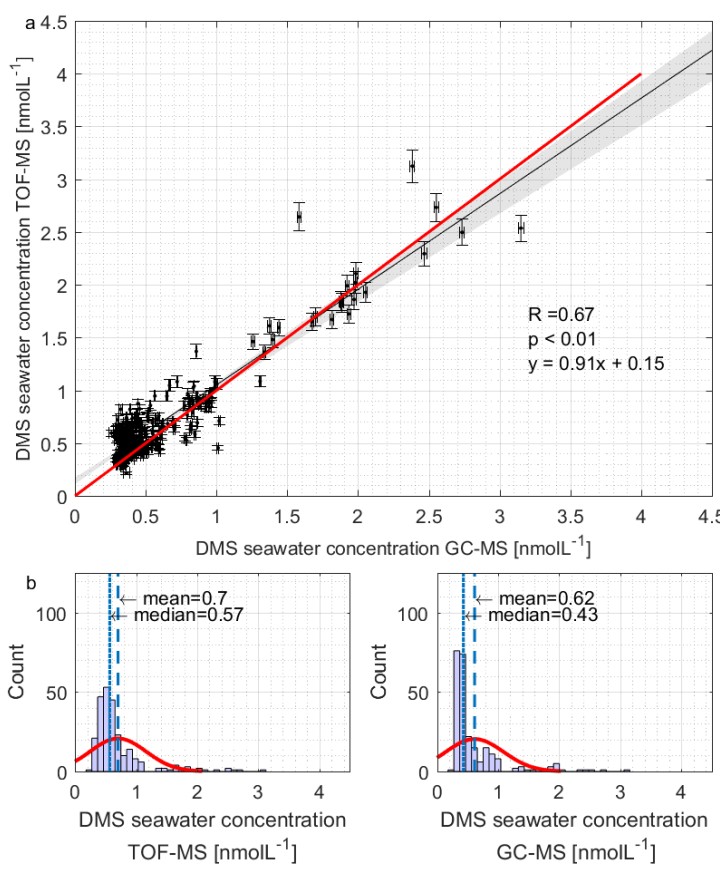

**Figure 2: The correlation (a) and Gaussian distribution (b) analysis of DMS concentrations measured by GC-MS (right) and TOF-MS (left). TOF-MS data were matched to GC-MS data within a +/- 5 min timing interval. The black line and red line in (a) denote the regression line and the 1:1 line, respectively. The red line in (b) denotes the density curve.**

**3.2 Environmental characterization during SCALE**

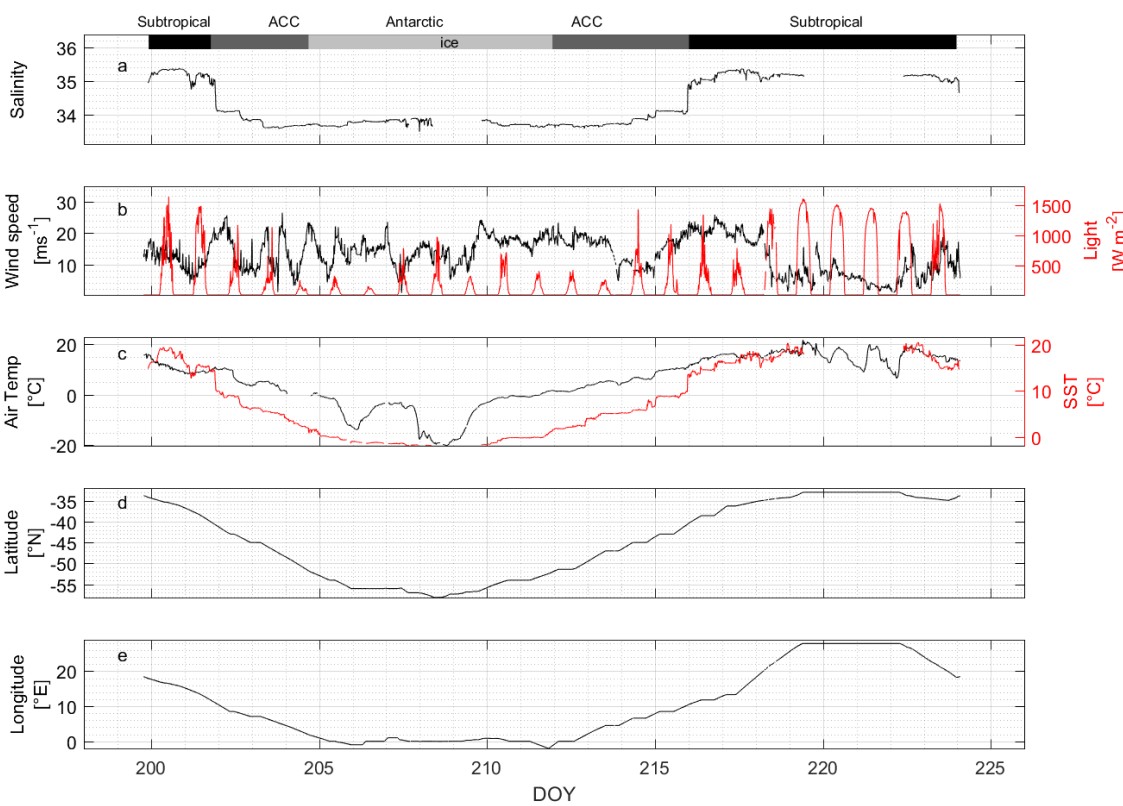

**Figure 3: Auxiliary parameters - in situ measurements of salinity (a), wind speed at 18 m (b, black), light levels (b, red), air temperature (c, black) and SST (c, red), cruise track latitude (d), and cruise track longitude (e) averaged over 10 minutes. The bars across the top of the figure denote the different hydrographic regions discussed throughout the text.**

The cruise started in the Atlantic Ocean, at Cape Town, and sailed, crossing different fronts, to the Southern Ocean
ice zone before returning along almost the same track, via East London, back to Cape Town (Figure 1). According to
the ratio of SST and salinity along the track, three different regions were identified, namely the subtropical, Antarctic
Circumpolar Current (ACC), and Antarctic regions (Figure 3; see supplemental information for more details on the
defining characteristics, Figure S2).  At DOY 201.91 and 216.01, we observed a change between the ratios greater
than 0.05, but did not find such a strong change at other observation points, thus distinguishing the Subtropical and
ACC regions. For the Antarctic region, we found that the ratio of change intensity was not as strong as between
Subtropical and ACC, but did occur (a change of 0.04). From this ratio, we distinguish the two regions of ACC and
Antarctic. Luis and Lotlikar (2021) report that the range of SST in the ACC region is from 1.8°C to 9°C, and that of
salinity is from 33.85 to 34.8; the SST of the Antarctic area is less than 1.5°C, and the salinity is less than 34. These
ranges are consistent with our SST-salinity-ratio defined regions. Unfortunately, in the ice area (DOY 208.4 - 209.8),



surface seawater was not collected, because the inlet of the underway pump was blocked by the sea ice. The underway pump was shut down also when the ship docked at the port of East London (DOY 219.4 - 222.4). The lowest air temperature occurred at DOY 208, which was approximately -20°C (Figure 3c). During the cruise, wind speed (Figure 3b) was on average $15.0 \pm 2.5$ m s$^{-1}$. We experienced higher wind speeds between DOY 210 and 214 and DOY 216 -
217, averaging over 20 m s$^{-1}$. As the ship approached the coast, winds slowed down and air temperatures rose.

### 3.3 Distribution of dissolved DMS and related compounds

**Table 1: Literature review of published field studies of DMS concentrations and mixing ratios (seawater, air) during wintertime (SO: Southern Ocean; O: Open Ocean; I: Island; C: Coastal).**

| Reference | Area | Water Type | DMSwater nmol L$^{-1}$ | | DMSair pptv | |
|---|---|---|---|---|---|---|
| | | | Mean | Range | Mean | Range |
| Lee and De Mora (1996) | New Zealand | O/C | 1.82 | 1.51 - 2.82 | | |
| Nguyen et al. (1992) | Amsterdam Island, India Ocean | C/I | 0.20 | | 13.20 | |
| Gibson et al. (1988) | Antarctic | O/C | 1.38 | 1.11 - 1.64 | | |
| Nguyen et al. (1990) | southern Indian Ocean | O/C | 2.51 | 0.30 - 2.01 | 58 | 34-274 |
| Akademik Korolev, 1987, unpublished | Indian and Pacific Oceans | O | 0.75 | 0.31 - 1.25 | 1.25 | |
| Marion Dufresne, 1998, unpublished | Indian Ocean | O | 0.96 | 0.37 - 2.01 | 2.01 | |
| This study | SO / whole cruise | O | $1.03 \pm 0.98$ | 0.26 - 5.18 | $28.80 \pm 12.49$ | 0.06 - 88.68 |
| | SO / subtropical region | | $1.86 \pm 1.05$ | 0.45 -5.18 | $23.25 \pm 7.16$ | 7.49 - 35.71 |
| | SO / ACC region | | $0.48 \pm 0.15$ | 0.30 - 1.00 | $29.16 \pm 9.41$ | 0.06 - 58.34 |
| | SO / Antarctic region | | $0.36 \pm 0.04$ | 0.26 - 0.47 | $31.40 \pm 16.77$ | 0.06 - 88.68 |

During the campaign, the mean sea surface concentration of DMS was $1.03 \pm 0.98$ nmol L$^{-1}$ using the GC-MS (0.75 $\pm$ 0.52 nmol L$^{-1}$, TOF), ranging from 0.26 nmol L$^{-1}$ to 5.18 nmol L$^{-1}$ (0.21 - 3.96 nmol L$^{-1}$, TOF, Figure 4a). The concentrations measured during SCALE are comparable to those from previous winter measurements (Table 1). Measurements made at lower latitudes appear to be consistently higher than those made at higher latitudes. Surface seawater DMS levels in the Southern Ocean during winter were much lower than those measured in spring (e.g., 0.4
- 7.9 nmol L$^{-1}$ and 3 - 40 nmol L$^{-1}$ (Curran and Jones, 2000; Kiene et al., 2007)), summer (e.g., 0.6 - 30 nmol L$^{-1}$ (Tortell and Long, 2009)), and autumn (e.g., 0.7-3.3 nmol L$^{-1}$ and not detected (nd) - 27.9 nmol L$^{-1}$ (Wohl et al., 2020; Yang et al., 2011; Zhang et al., 2020)). The average concentration in the subtropical region was the highest during the entire cruise, which was $1.86 \pm 1.05$ nmol L$^{-1}$, while the average concentrations in the ACC and Antarctic regions were significantly lower, $0.48 \pm 0.15$ and $0.36 \pm 0.04$ nmol L$^{-1}$, respectively. The southern and northern ACC transects
and the Antarctic regions presented a similar concentration range between 0.26 and 1.00 nmol L$^{-1}$, however, over the two observation periods of the subtropical region, the concentration ranges of DMS were not similar. The subtropical region along the first transect (southwards) exhibited lower concentrations than same area of the second transect (northwards). This difference could be due to the duration of time spent sampling at the coast: at the beginning of the cruise, the coastal zone was left behind relatively quickly, while on the return trip, a greater period of time was spent
sampling the near-shore waters. Fluctuations in DMS concentrations were visible where the subtropical region and





the ACC circulation met, along with SST and salinity changes. Finally, in order to understand uncertainties induced by sparse sampling in winter, we compare our data to the Lana climatology (Lana et al., 2011). We find that our measured DMS concentrations over the cruise track area are lower than the climatological concentrations (Figure 5), particularly in the ACC circulation region.

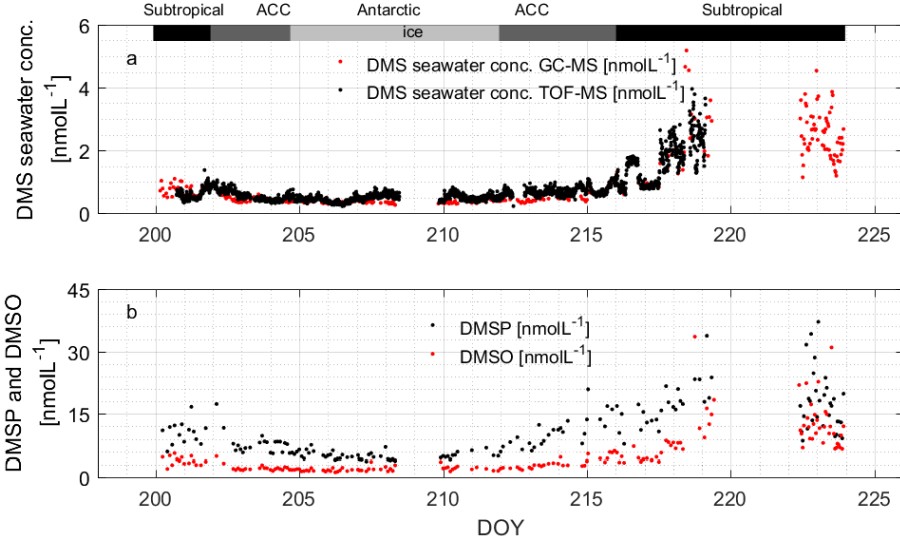


**Figure 4: (a) Measured DMS concentrations in seawater using both instruments. (b) Measured DMSP and DMSO concentrations in the sea surface using the GC-MS. The bars across the top of the figure denote the different hydrographic regions discussed throughout the text.**

The average sea surface concentration of DMSP during the observation period was $11.26 \pm 6.98$ nmol L$^{-1}$, and the

distribution range was 3.73 - 40.27 nmol L$^{-1}$ (Figure 4b). By comparing to previous research, it can be seen that the concentration during the entire observation period is lower than most of the existing literature values, likely because biological activity in winter is low, resulting in the lowest concentration of DMSP (Curran et al., 1998; Jones et al., 1998; Kiene et al., 2007). The only values that are similarly low are also from the winter season (Cerqueira and Pio, 1999). The highest concentrations were found in the subtropical region (average: $16.48 \pm 7.08$ nmol L$^{-1}$). Average

concentrations in the ACC and Antarctic regions were $9.00 \pm 3.44$ and $5.19 \pm 0.94$ nmol L$^{-1}$, respectively.

The average sea surface concentration of DMSO during the observation period was $5.41 \pm 5.31$ nmol L$^{-1}$, and the distribution range was 1.18 - 33.56 nmol L$^{-1}$ (Figure 4b). The subtropical region had the highest average concentration of $9.28 \pm 6.15$ nmol L$^{-1}$, and the average concentrations of the ACC and Antarctic regions were similar, $2.74 \pm 1.17$

and $1.90 \pm 0.52$ nmol L$^{-1}$, respectively. Kiene et al. (2007) measured the concentration of dissolved DMSO in the Southern Ocean in the summer of 2009, and the concentration range (1 - 55 nmol L$^{-1}$) was higher than the concentration of total DMSO in this study. Our study area was not subject to human interference, and the biological activity, as well as light levels, in winter are low, therefore our wintertime measured concentrations can be regarded as the lowest background value of the area.

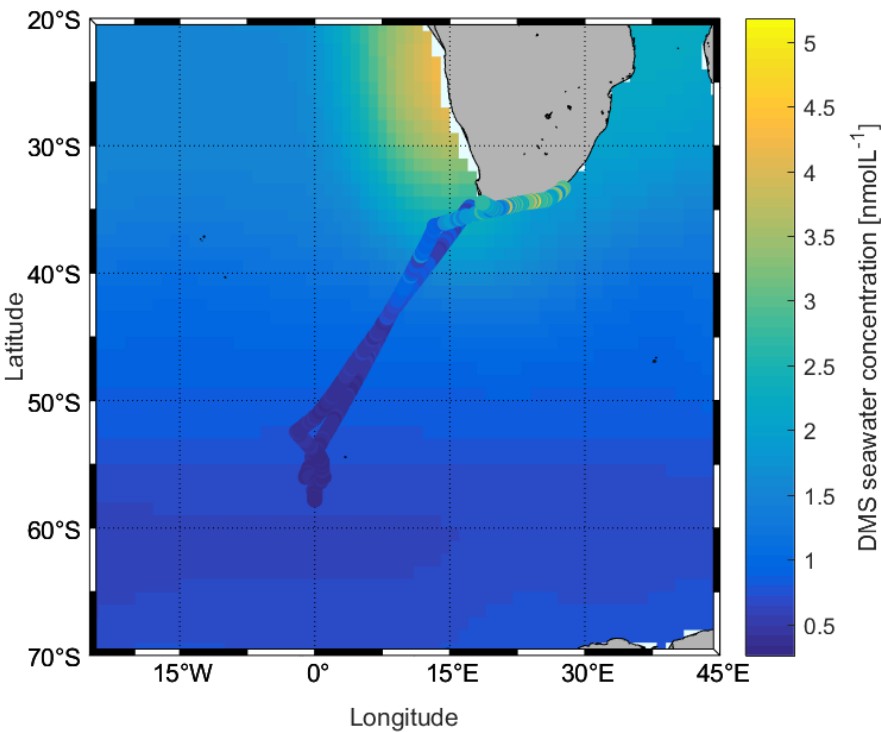


**Figure 5: Comparison between measured DMS concentrations (cruise track trace) and those in the Lana et al. (2011) climatology (background) for August.**

### 3.4 Relationships between sulphur compounds

Correlation analysis of DMS and DMSP/O in different circulation regions (Figure 6) shows that the regional
relationships are different. In the subtropical area, the slopes of DMS against DMSP/O are similar, with P values less
than 0.01 and $R^2$ values higher than 0.5. Studies have shown that DMSO can be a source of DMS in areas with strong
sunlight (Zindler et al., 2015). Therefore, in the subtropical area, the source of DMS relates to both DMSP and DMSO.
The slope of DMSP against DMSO is 0.64, with a P value less than 0.01 and $R^2 = 0.6$ (Figure 6c). This information,
along with the similar concentration range for both DMSO and DMSP (ca. 40 nmol $L^{-1}$), indicates a tight coupling
between DMSP and DMSO. Overall, the positive correlation between DMS, DMSP and DMSO indicates that there
is strong cycling between the three in this region.

We further split the subtropical data into coastal (blue dots, Figure 6a - c) and open ocean (red dots, Figure 6a - c), as
we see that there is a distinct concentration difference in these subregions. We again analyzed the relationship between
DMS, DMSP and DMSO in these two areas and we find that the concentration of DMS, DMSP and DMSO is higher
at the coasts than in the open ocean. The $R^2$ values between DMS and DMSP are 0.81 and 0.38 in open seas and near
coast areas, and the slopes are 8.14 and 7.13, respectively. The P values are less than 0.01 for both subregions. This
result shows that closer to the subtropical coastal area the DMSP has a positive effect on DMS, with a lesser impact





in the open sea. While it has been shown that more microbial DMSP cleavage to DMS occurs at the coast (Zubkov et al., 2002), the relationship between the two compounds may be confounded by DMS interconversion with DMSO (as

discussed below). The $R^2$ values between DMS and DMSO in the open ocean and the coastal regions are 0.77 and 0.61, the P values are less than 0.01, and the slopes are 3.61 and 6.02, respectively. The obvious difference in correlation and slope indicates that the relationship is different, possibly because the coastal region may contain more photosensitizers (c/fdom) promoting increased photochemical cycling between the two compounds (Mopper and Kieber, 2002). Comparing the relationship between DMSP and DMSO, we find that in the near coast area, the $R^2$

value between DMSP and DMSO is 0.50, while in the open sea it is 0.76, the P values are less than 0.01, and the slopes are 0.49 and 0.37, respectively. This may indicate that DMSO is higher at the coast, not because of direct algal production of DMSO along with DMSP, but because there is more DMSP to DMS microbial cleavage (Figure 6a) and strong cycling between DMS and DMSO due to enhanced photochemistry (Figure 6b) - where DMSP is high because of greater biological productivity (Hatton et al., 2004; Hatton et al., 2012; Stefels et al., 2007).


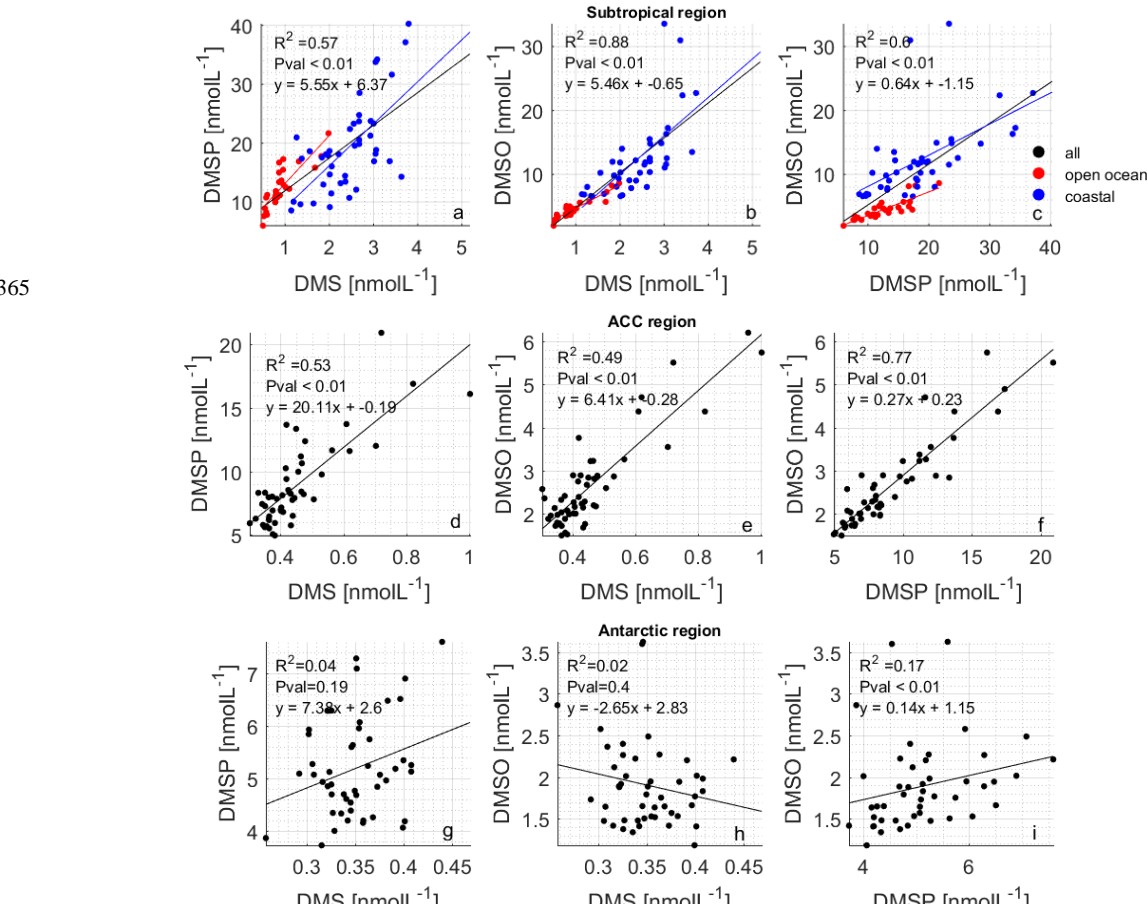


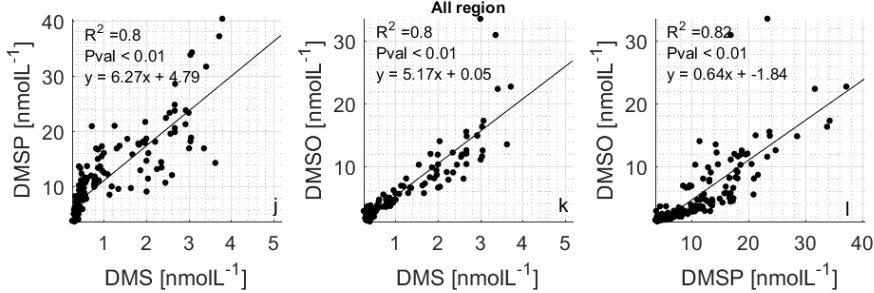

**Figure 6: Correlations between measured DMSP and DMS (the left column of the figure), DMSO and DMS (the middle column of the figure) and DMSO and DMSP (the right column of the figure); a-c) subtropical region (black points); d-f) ACC region; g-i) Antarctic region; and j-l) entire cruise. The red points in a-c are data from open ocean in the subtropical area and the blue points are data from the subtropical area near the coast. The results of the correlation analysis between DMS and DMSP are: $R^2$=0.81, Pval < 0.01, y = 8.14x + 5.01 for open ocean waters; $R^2$=0.38, Pval < 0.01, y = 7.13x + 1.9 for coastal waters. The results of the correlation analysis between DMS and DMSO are: $R^2$=0.77, Pval < 0.01, y = 3.61x + 1.12 for open ocean waters; $R^2$=0.61, Pval < 0.01, y = 6.02x + -2.04 for coastal waters. The results of the correlation analysis between DMSP and DMSO are: $R^2$=0.76, Pval < 0.01, y = 0.37x + -0.11 for open ocean waters; $R^2$=0.5, Pval < 0.01, y = 0.49x + 3.26 for coastal waters.**

The concentration range of DMS and DMSP/O in the ACC region is reduced by half or even more than that of the subtropical area. In addition, it can be seen that the slope (20.1) between DMS and DMSP is significantly higher than that between DMS and DMSO or DMSP and DMSO. The P value is less than 0.01, and the $R^2$ value is greater than 0.5, indicating that DMSP has a leading role in the generation of DMS. The higher slope means that more DMSP is required in the ACC region to produce DMS compared to the subtropical region. This again may indicate different microbial pathways leading to higher DMS production in the subtropical area.

In the Antarctic circulation area, the P values between DMS and DMSP, DMS and DMSO are all greater than 0.01, indicating that the sources of DMSP and DMSO are not connected. Although the P value between DMSP and DMSO is less than 0.01, the $R^2$ value is only 0.17, supporting the idea that the cycling of the compounds is relatively decoupled (Figure 6 Antarctic region). There appears to be little to no biological activity there. Additionally, due to the low solar radiation dose during winter in the Southern Ocean (Figure 3g), little DMSO production via photoreaction from DMS is expected (Vallina and Simo, 2007).

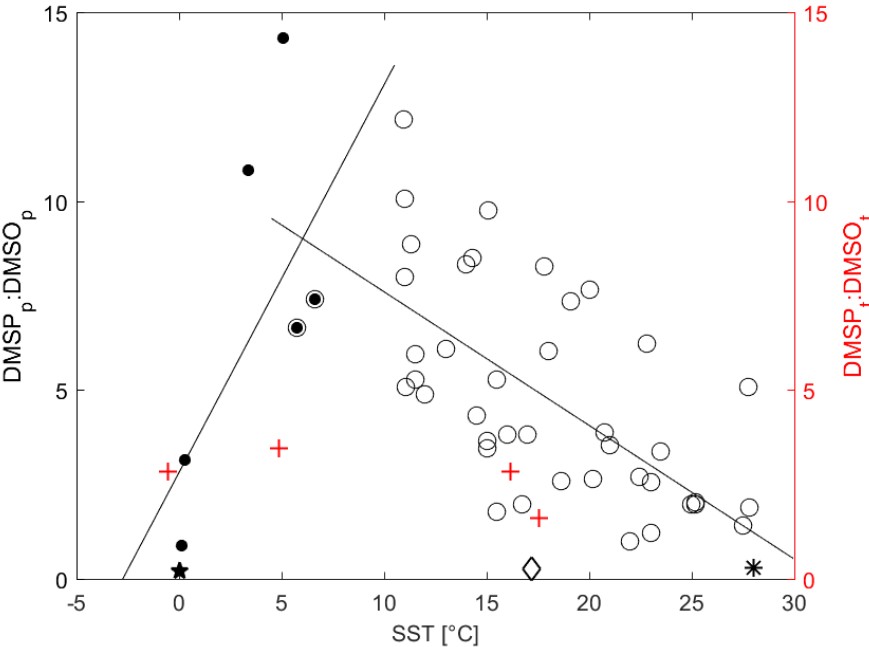


**Figure 7: Average DMSPt:DMSOt vs SST (red) and average DMSPp:DMSOp vs SST (black). Mean ratios for individual campaigns are recalculated from the data listed in Simó and Vila-Costa (2006). We added data points consisting of the mean DMSPp:DMSOp and SST (given in parenthesis) from the East China Sea (0.27, 17.2 °C, open diamond) (Yang and Yang, 2011), the northern Baffin Bay (0.20, estimated 0 °C; closed pentagram) (Bouillon et al., 2002) and the western Pacific Ocean (0.22, 28 °C, open asterisk) (Zindler et al., 2013). The linear correlations are y = -0.35x + 11.13 (R² = 0.45, open circles) and y = 1.03 x + 2.82 (R² = 0.57, solid circles). Our Southern Ocean, ACC, and subtropical open ocean and coastal areas are the red plus signs.**

Simo et al. (2000) found that the relationship between DMSPp:DMSOp and SST points to the presence of coccolith

blooms that lead to high levels of DMSP. Simó and Vila-Costa (2006) found that the particulate DMSP (DMSPp) and

DMSO (DMSOp) ratio has a negative correlation with SST and latitude. Zindler et al. (2013) found that the trend

changes sign at temperatures below 5°C. The temperature in our observation area varies widely, so it is a good dataset

for determining if this change in relationship with SST is robust. Unfortunately, we did not measure DMSPp and

DMSOp, so we compare total DMSP:DMSO with SST (Figure 7). Indeed, we find that our data corroborates both

studies (Figure 7, red plus signs and Figure S3), with an increasing relationship at low temperatures until about 5°C

and then a decreasing relationship in warmer waters. The relationship between DMSPt and DMS to SST over the

entire cruise was also investigated (Figure S3). The pattern observed in DMSPt:DMSOt associated with SST above

5-10°C is likely due to the variation of DMSO production rate associated with the change of solar radiation dose. High

DMSO production rates coupled to high DMSP degradation rates under high SST conditions causes a decline in the

observed ratio with temperature. The opposite is true in colder waters with corresponding low light levels (Figure 3),

leading to an increase in the ratio with temperature until around 5°C. As is discussed in the supplemental material in

more detail, DMSPt:DMS follows a similar trend as DMSPt:DMSO to SST, which may be due to decreasing DMSP

production with temperature and increasing DMSP to DMS microbial cleavage (Stefels et al., 2007; Yoch, 2002).



### 3.5 DMS atmospheric mixing ratios and fluxes

The average DMS mixing ratio in the boundary layer throughout the observation period was $28.80 \pm 12.49$ (0.06-88.68) pptv. The averages for each region were $23.25 \pm 7.16$ (7.49 - 35.71) pptv, $29.16 \pm 9.41$ (0.06 - 58.34) pptv, and $31.40 \pm 16.77$ (0.06 - 88.68) pptv in the subtropical, ACC, and Antarctic regions, respectively. These values fall within the range of previously reported winter atmospheric mixing ratios over the Southern Ocean (Table 1), which are lower than those reported for spring (nd-755 pptv, (Inomata et al., 2006) and autumn (nd-3900 pptv, (Zhang et al., 2020).

The temporal trends of atmospheric DMS mixing ratios during our research campaign were different from those observed in seawater (Figure 8c). For example, the highest atmospheric concentrations of DMS were found in the Antarctic region, where seawater concentrations were the lowest. The likely reasons for this include lower atmospheric photochemical reaction rates, a lower boundary layer, and DMS release from ice (Koga et al., 2014) . The lower reaction rates result in an increased lifetime of DMS and a build-up of DMS in the boundary layer. Low sea surface

temperatures create a lower atmospheric boundary layer during winter, which aids in DMS build-up. Finally, it has been shown that when research vessels travel in the ice area and crush the ice, higher concentrations of DMS can be released from the gap between the ice and the sea surface, which also increases the concentration of DMS in the air (Koga et al., 2014).

DMS fluxes were calculated using two different gas exchange coefficient parameterizations from Zavarsky et al. (2018, Z18) and Nightingale et al. (2000, N00) (Figure 8b). The Z18 parameterization is based on direct flux measurements of DMS, while the N00 values are from dual tracer studies of $^3$He/SF$_6$. For our purposes, the Z18 parameterization is preferred, but in order to compare with the Lana climatology N00 is used as well. It can be seen from the results that $k$Z18 ($17.78 \pm 7.30$, 0.99 - 43.48 cm hr$^{-1}$) is lower than $k$N00 ($29.78 \pm 20.14$, 0.12 - 131.52 cm hr$^{-1}$) over the windspeed

range observed during SCALE. Values of $k$Z18 are $20.77 \pm 9.18$ (2.32 - 43.49), $17.27 \pm 5.72$ (1.10 - 38.40) $14.93 \pm 4.00$ (0.99 - 31.22) cm hr$^{-1}$ in the subtropical, ACC, and the Antarctic regions, respectively. The values of $k$N00 are $34.18 \pm 26.29$ (0.37 - 123.90), $29.67 \pm 16.56$ (0.12 - 131.51) and $25.50 \pm 11.92$ (0.12 - 100.83) cm hr$^{-1}$, respectively. Especially in areas with high wind speeds (DOY 216 - 218), $k$N00 is significantly higher than $k$Z18. The difference lies in the wind speed dependency of the two parameterizations: Z18 is linear, while N00 has a quadratic term. This

difference in functional form is, likely, because the solubilities of the dual tracer gases and DMS are different, which could lead to discrepancies at high wind speeds where bubble-mediated gas transfer is important (i.e. more soluble gases, such as DMS, have a lower bubble-mediated gas exchange potential). Therefore, the N00 parameterization may not be applicable to DMS fluxes at high winds. However, the difference between $k$Z18 and $k$N00 data is not significant from DOY 219 to 225, which corresponds to the wind speeds below 10 m s$^{-1}$ (Figure 8a).

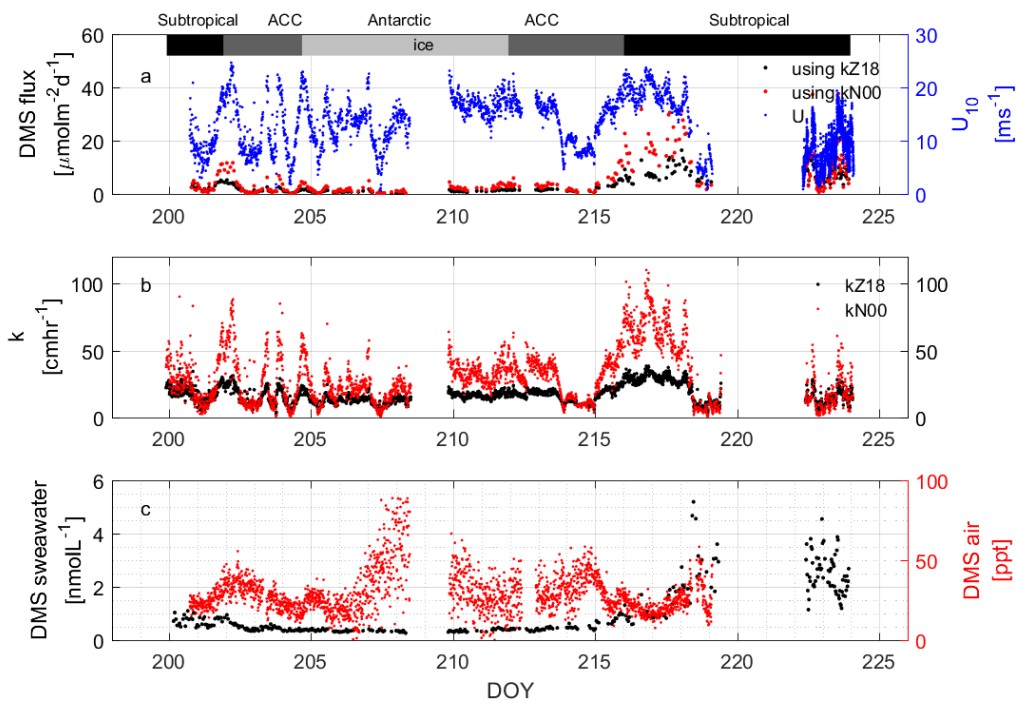


**Figure 8: Calculated DMS fluxes (a) that depend on the indicated wind speeds (U from 10 m) using two different *k* values (b). The measured water and air values that were used to compute the concentration difference are shown in c. The bars across the top of the figure denote the different hydrographic regions discussed throughout the text.**

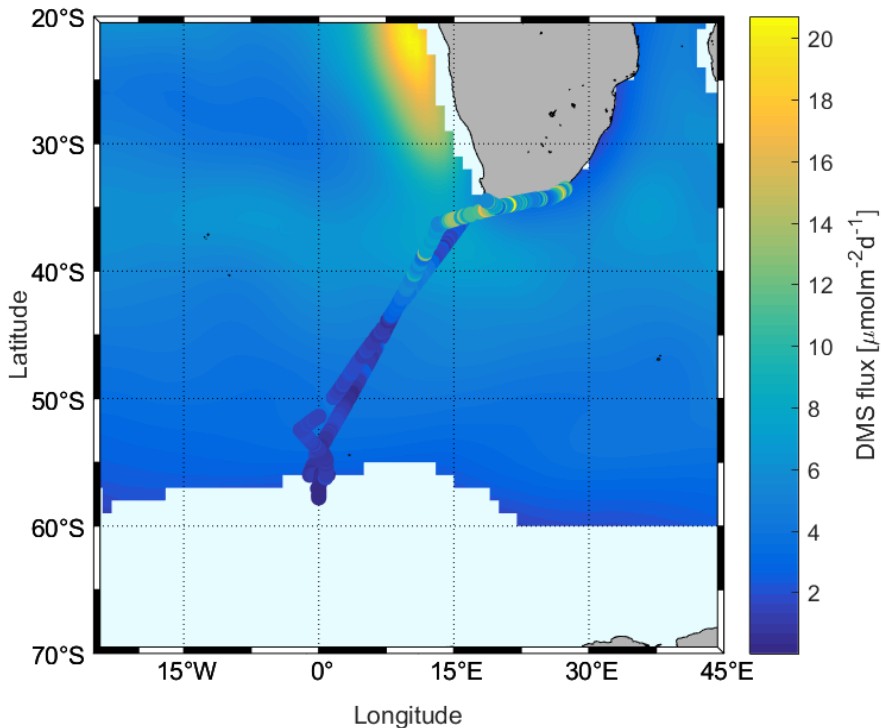

**Figure 9: Comparison between calculated fluxes using N00 during the SCALE cruise (cruise track trace) and those in the Lana et al. (2011) climatology (background) for August.**

The average flux calculated using $k$Z18 is $4.04 \pm 4.12$ μmol m$^{-2}$ d$^{-1}$, and the range is 0.02 to 22.03 μmol m$^{-2}$ d$^{-1}$. The average calculated flux using $k$N00 is $6.10 \pm 7.08$ μmol m$^{-2}$ d$^{-1}$, and the range is 0.04 to 37.12 μmol m$^{-2}$ d$^{-1}$. In the subtropical, ACC and Antarctic regions, the average fluxes (ranges) calculated using $k$Z18 are $7.63 \pm 4.29$ (1.00 -

22.03), $2.00 \pm 1.33$ (0.21 - 6.12) and $1.07 \pm 0.51$ (0.02 - 2.02) μmol m$^{-2}$ d$^{-1}$, respectively, while the average fluxes (ranges) calculated using $k$N00 are $10.88 \pm 8.71$ (0.67 - 37.62), $3.53 \pm 3.05$ (0.06 - 12.60) and $1.95 \pm 1.14$ (0.04 - 5.06) μmol m$^{-2}$ d$^{-1}$. For areas with high concentrations of DMS in the water and high wind speeds, the computed fluxes using $k$N00 can be twice as much as those using $k$Z18 (DOY 216 - 218). In areas with high wind speed and low concentrations of DMS in the water, the effect on the calculated flux is not as obvious, despite the difference in $k$

values (e.g., DOY 207), and both computed fluxes remain low over the region. We do not observe any influence of atmospheric mixing ratios on the computed flux. Overall, we calculate that in high wind speed (> 20 ms$^{-1}$) areas, the different parameterizations have a large impact on the computed fluxes and the $k$ value should be considered more carefully in climatologies to avoid errors in the flux calculation.

We also compared our calculated flux results with the Lana climatology and it can be seen from Figure 9 that there are clear differences. The climatology shows lower results than those calculated from our observations in the





subtropical region, but higher values in the ACC region, at high latitudes (> 43°S). The differences are due to differences in seawater concentrations used to calculate the fluxes, where our observations were slightly higher than in Lana et al. (2011) for parts of the subtropics and lower than in Lana et al. (2011) in the ACC. The subtropical region

between 35 and 40°S, however, presents unexpected disagreement between the datasets, where the SCALE observations were similar to the climatology, but the SCALE fluxes are obviously higher. This is due to the differences in wind speed encountered during our cruise in comparison to the monthly mean winds used in Lana et al. (2011). Finally, within the ACC region, the flux of DMS decreases rapidly, unlike the pattern displayed in the Lana climatology.

**3.6 Dissolved isoprene distribution**

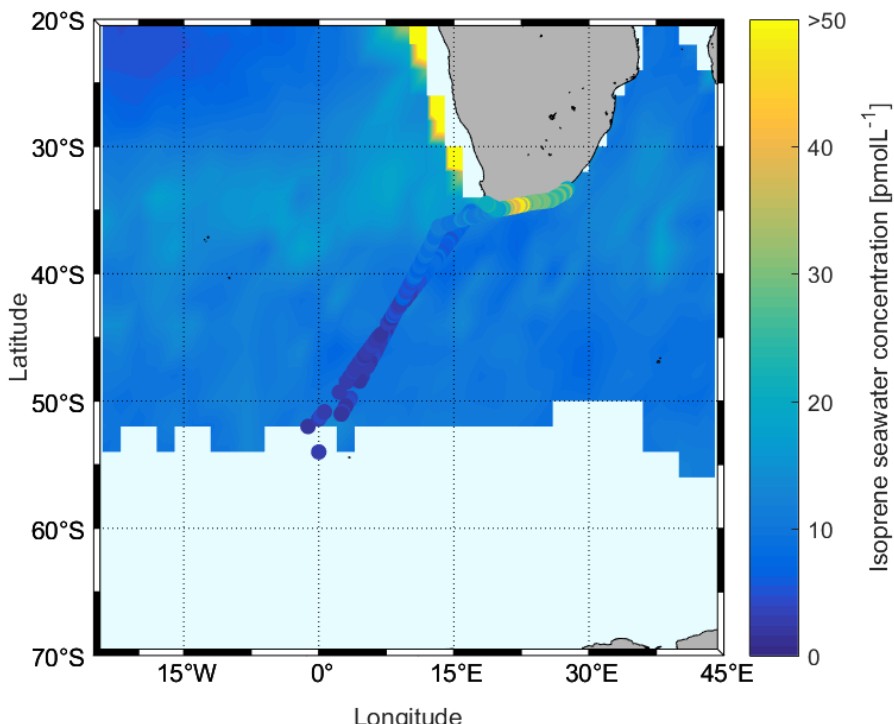

**Figure 10: Comparison between measured sea surface isoprene concentrations during the SCALE cruise (cruise track trace) and those computed using a satellite-based model (Booge et al., 2016) for August 2019.**

**Table 2: Isoprene seawater concentration observations in the Southern Ocean (SO).**

| Reference | Area | Time | Isoprene water pmol L⁻¹ | |
|---|---|---|---|---|
| | | | Mean | Range |
| Wohl et al. (2020) | SO | Autumn | 13.4 ± 6.3 | 5.0 - 50.0 |
| Rodríguez-Ros et al. (2020) | SO | Summer | 10.7 | 2.1 - 88.4 |
| Rodríguez-Ros et al. (2020) | SO and Weddell Sea | Summer | 22.4 | 1.6 - 93.5 |





| Rodríguez-Ros et al. (2020) | Southwestern Atlantic Self | Autumn | 25.3 | 12.0 - 49.5 |
|---|---|---|---|---|
| Kameyama, et al. (2014) | SO | Summer | 78.7 | 0.2 - 348 |
| This study | SO / whole cruise | Winter | 14.46 ± 12.23 | nd - 54.00 |
| | SO / subtropical region | | 20.23 ± 11.58 | 4.42 - 54.00 |
| | SO / ACC region | | 3.76 ± 1.46 | 2.06 - 7.65 |
| | SO / Antarctic region | | 2.66 | nd-2.66 |

In our study, we observed that the isoprene concentrations ranged from nd to 54.00 pmol L$^{-1}$, and the average was 14.46 ± 12.23 pmol L$^{-1}$. These concentrations are within the range of published values (Table 2). Although our observation season is winter, on average, our measurements are not lower than those aboard the Antarctic Circumnavigation Expedition observed during December 2016 - March 2017 on the R/V Akademik Treshnikov (Rodríguez-Ros et al., 2020) and similar to the research cruise ANDREXII during autumn (February – April 2020)

(Wohl et al., 2020). When we examine the different regions, we find the lowest isoprene concentrations in the Antarctic region (1 datapoint only - 2.66 pmol L$^{-1}$), followed by the ACC region (average: 3.76 ± 1.46 pmol L$^{-1}$, range: 2.06 - 7.65 pmol L$^{-1}$), and highest concentrations in the subtropical region (average: 20.23 ± 11.58 pmol L$^{-1}$, range: 4.43 - 54.00 pmol L$^{-1}$). When we compare these regional values to previously published isoprene concentrations, it is obvious that wintertime concentrations are lower than other seasons (ACC vs. the R/V Akademik Treshnikov

expedition). Results from a comparison with modelled surface isoprene concentrations (Booge et al., 2016) show that measured winter time isoprene concentrations in the ACC and Antarctic regions are lower than expected (Figure 10). Modelled isoprene concentrations in the open ocean subtropical region agree with our measurements, whereas the model seems to underestimate surface isoprene concentrations in coastal areas.

### 3.7 Isoprene air/sea fluxes

We calculated isoprene using two different gas exchange coefficient parameterizations, Wanninkhof (1992, W92) and Wanninkhof (2014, W14) (Figure 11). We recommend using W14 to calculate fluxes of rather insoluble gases, as it is the updated version of W92 (based on later results). However, we use the W92 parameterization to compare to the isoprene flux model results from Booge et al. (2016). Generally, it can be seen that the $k$ value has a large influence on the calculated fluxes. The values of $k$W92 are the highest (average: 30.43 ± 20.74 cm hr$^{-1}$, range: 0.03 - 143.18 cm

hr$^{-1}$), but $k$W14 are within 17.64% (average: 24.64 ± 16.80 cm hr$^{-1}$, range: 0.03 - 115.93 cm hr$^{-1}$). Values for $k$W14 in the subtropical, ACC, and Antarctic region are 26.88 ± 21.59, 25,01 ± 14.29, 22.56 ± 10.92 cm hr$^{-1}$, respectively. Values for $k$W92 are 33.20 ± 26.66, 30.88 ± 17.64, 27.86 ± 13.48 cm hr$^{-1}$, respectively. The differences between the two parameterizations are related to the magnitude of the coefficient, not the functional form of the wind speed dependence.






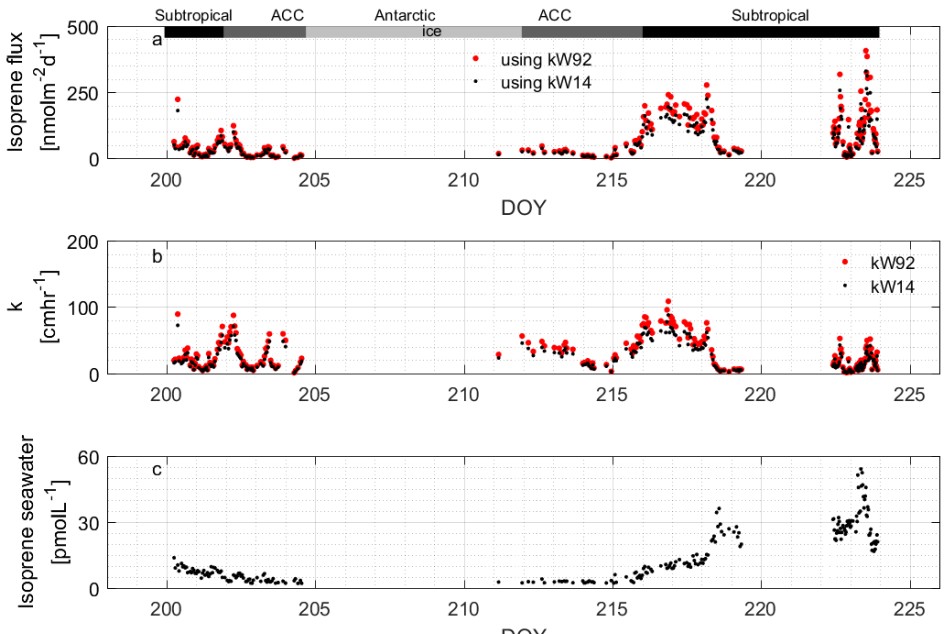

**Figure 11: Calculated isoprene fluxes (a) using two different *k* values (b). The measured water values that were used to compute the fluxes are shown in (c). The bars across the top of the figure denote the different hydrographic regions discussed throughout the text.**

The computed isoprene flux ranged from nd to 407.05 nmol $m^{-2}$ $d^{-1}$, with an average of 80.55 ± 78.57 nmol $m^{-2}$ $d^{-1}$. The average isoprene fluxes (ranges) in the subtropical, ACC, and Antarctic regions were 107.36 ± 84.30 (4.66 - 407.05) nmol $m^{-2}$ $d^{-1}$, 31.23 ± 26.85 (0.38 - 123.29) nmol $m^{-2}$ $d^{-1}$, and 18.03 nmol $m^{-2}$ $d^{-1}$ (one data point in Antarctic region), respectively. We observed that the isoprene fluxes can change rapidly and that changes in wind speed are the main factor driving the flux of isoprene, which is rather insoluble. This can be seen comparing isoprene concentrations

and resulting fluxes during two time periods DOY 200 - 202 and DOY 216 - 218 when passing through the same region (subtropical open ocean region, Figure 11). Isoprene concentrations were similar (8.01 ± 1.87 and 10.37 ± 1.97 pmol $L^{-1}$), but the fluxes from DOY 216 - 218 are on average 3.9 times higher than during time period DOY 200 - 202. Isoprene fluxes fluctuated between 5.30 and 463.24 nmol $m^{-2}$ $d^{-1}$ in the coastal area and were influenced by varying surface isoprene concentrations and wind speed. However, it can be seen in the back trajectories (Figure 1,

right) that air masses from land reached coastal waters within a 24-hour time period, which renders the assumption that the isoprene air mixing ratio of 0 unlikely. Thus, the flux values computed at the coast should be treated as upper limits.

Finally, we compared the results with Booge's model (background, Figure 12). It can be seen that the overall model-

based flux range is similar to the calculated fluxes using actual observations (Figure 12). However, when comparing


individual regions, we see that in the ACC and Antarctic region fluxes based on actual surface concentration measurements are lower than predicted by the model. This is also true for some parts of the subtropical region, but variations are much higher, which also results in much higher isoprene fluxes than the Booge model, although the pattern is patchy. In addition, in the northern ACC as well as in the subtropical region, the outbound and return travel

areas are close together, but the fluxes span a wide range. The background value of the model is closer to the flux calculations on the outgoing trip, and the return trip is much higher than the outgoing trip, which is due to the change in wind speed.

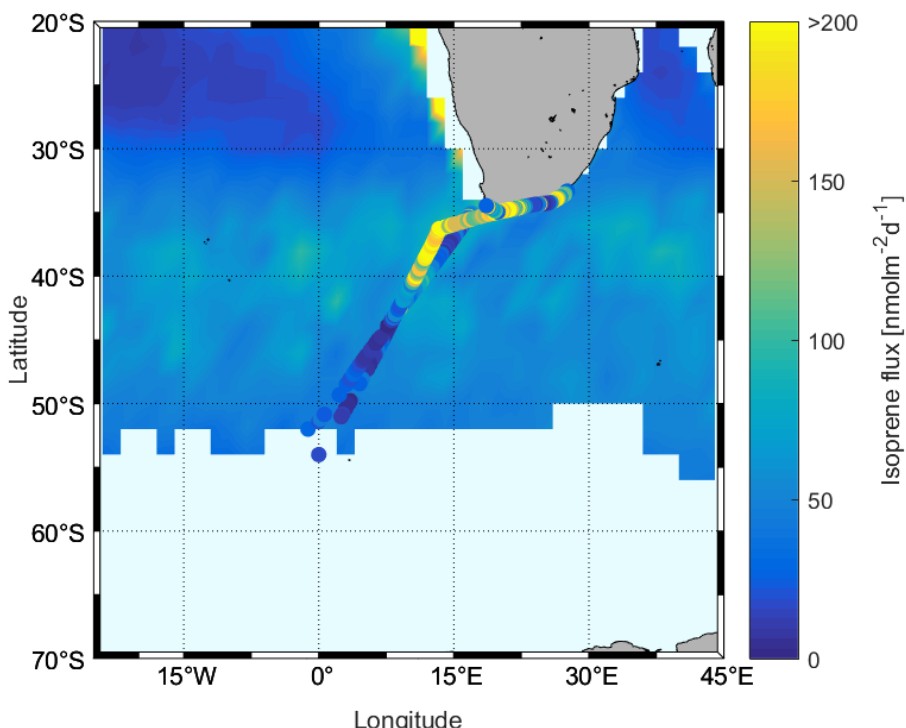

**Figure 12: Comparison of calculated fluxes using W92 during the SCALE cruise (cruise track trace) and model-based fluxes using W92 from Booge et al. (2016) (background – monthly mean values of August 2019).**

## 4 Conclusions and outlook

DMS, DMSP, DMSO, and isoprene were observed in the surface waters (and DMS in the air) of the Southern Ocean
during winter, where this type of data is severely limited. We found that all compounds in seawater showed a decreasing trend with latitude, while DMS in the atmosphere showed a maximum at high latitudes. The relationships between DMS, DMSP and DMSO in distinct regions along the cruise track indicate that different processes are at



work: there is no correlation among them in the Antarctic region, but positive correlation in lower latitude regions. Especially in the subtropical regions, the different results in the coastal and open sea reflect the complex cycling between the three compounds, likely due to the influence of temperature and light. We found low DMS fluxes during the Southern Ocean winter season. The calculated DMS fluxes using different k values suggest that previous studies might have overestimated the DMS flux. Results of the first published isoprene winter values show that measured concentrations are lower than those computed by satellite-based model data. Furthermore, the mismatch between model results and field measurements clearly indicates the need for more field data in this region and season in order to develop better parameterizations for models. Our results also emphasize the need for temporally and seasonally highly resolved models. Due to its insolubility, isoprene fluxes are highly influenced by the magnitude of wind speed. Although isoprene concentrations in the open ocean subtropical region were 10 pmol L$^{-1}$ or lower, during high wind speeds (DOY 216-218) fluxes reached 100-200 nmol m$^{-2}$ d$^{-1}$, which is significantly higher than coastal regions with low wind speed conditions and high oceanic isoprene concentrations (~30 pmol L$^{-1}$). These small-scale variabilities are currently not captured using e.g. monthly mean resolved models, which subsequently underestimates the influence of marine derived isoprene on atmospheric processes. For both DMS and isoprene, the choice of k parameterization and the influence of wind speed on computed fluxes is important and should be treated with care.

Data products, such as monthly climatologies, are extremely important tools. It is apparent that more data during the winter season is needed to create a robust set of climate-active trace gas climatologies and process-based models. Undersampling can cause large uncertainties in the climatologies, model output, and computed parameters (Jiang, 2020; Vandemark et al., 2011; Wiggert et al., 1994). Given that changes in wind speed can be short-term or have a long-term trend, it seems prudent to focus in the first place on obtaining robust maps of concentrations, perhaps at different time resolutions, that can be used with different wind products to compute fluxes. Furthermore, given that some trace gas databases (e.g. SOCAT, DMS PMEL) span decades, the data that is compiled into climatologies could reflect long-term trends that may need to be properly addressed. Finally, by comparing our measured field data with existing data products, some questions for future research emerge: How long-lived are fine spatial and temporal concentration/flux trends? How important (or misleading) are these finer scale observations for creating monthly climatologies of trace gases? How do annual changes in wind speed influence climatological flux calculations and how should this be reflected in new or updated climatologies? Future data gathering campaigns are needed to answer these questions and create optimized data products.

*Supplement.* The supplement related to this article is available on-line at:
Supplementary figures: Figure S1-S3.
Figure S1. DMS purge efficiency from <0°C to 20°C
Figure S2. Difference in ratio of temperature (SST) to salinity of surface waters during the SCALE cruise. The bars across the top of the figure denote the different hydrographic regions. ACC= Antarctic Circumpolar Current
Figure S3. DMSPt:DMSOt (left) and DMSPt:DMS (right) against SST over the entire SCALE cruise



*Author contributions.* CAM and DB designed project with help from LZ and MZ; LZ, MZ, DB performed measurements on board; LZ performed DMSP, DMSO measurements in lab; all authors interpreted the data and wrote the manuscript.

*Competing interests.* The contact author has declared that neither they nor their co-authors have any competing interests.

*Data availability.* DMS, DMSP, DMSO and isoprene data can be achieved through the author (will be published on Pangaea while in revision). The ship's data presented from the SCALE cruise is openly available at https://doi.org/10.5281/zenodo.6367852.: https://zenodo.org/communities/scale_south_africa/?page=1&size=20

*Acknowledgements.* The authors thank the captain and crew of S.A. Agulhas II as well as chief scientists Marcello Vichi (on board) and Sandy Thomalla (on shore) for their great support during the research cruise. Thanks to
Mohammed Zawad Reza for help with DMSP, DMSO measurements in GEOMAR lab. Additionally, we would like to thank PI Rafael Simo and Dr. Cathleen Schlundt for part of the raw data contributions to figure 7. The authors gratefully acknowledge the NOAA AirResources Laboratory (ARL) for the provision of the HYSPLIT transport and dispersion model used in this publication.

*Financial support.* This work was financed by the BMBF through grant 03F0782A (SO-TRASE), the China
Scholarship Council (grant no. 201606400066) and National Natural Science Foundation of China (NSFC) (42076226).

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
