# Peer review of "Winter season Southern Ocean distributions of climate-relevant"

_Biogeosciences, 2022_

## Author Comment (AC1)

We thank Reviewer #1 for his in-depth analysis and detailed comments on our work, allowing us to revise and improve the manuscript. All critical points raised have been considered and addressed/discussed as detailed in this document (blue). Every indication of lines and/or pages refer to the originally uploaded manuscript.

1. On page 2, line 65, '400-600 Tg C yr-1; (Arneth et al…', Delete, '('.

   Done.

2. On page 2, line 66, 'plays an important role in the chemistry there…', it is recommended to revise 'there' to 'locally'.

   Thanks for pointing that out. Changed to "locally".

3. On page 3, line 71, it is recommended to delete 'subsequently'.

   Deleted.

4. On page 5, for Figure 1(right), the cruise track is not clear. Please revise it.

   We revised Figure 1 by using a bold line for the cruise track.

[Figure]

5. On page 10, I suggest combining Figure 3(d) and (e).

   We combined figure 3 (d) and (e). It will look as follows in the manuscript:

[Figure]

6. The descriptions of isoprene are much shorter than those related to DMS in the manuscript. However, the data of surface water isoprene during the winter season of the Southern Ocean was possibly reported for the first time. It looks like the isoprene is not important.

Isoprene is not unimportant, but earlier studies determined that seawater isoprene concentrations were low and did not further study the compound. Subsequent studies have shown that marine isoprene may have an impact on climate (e.g. Shaw et al., 2010, Bonsang et al., 1992), which has led to renewed interest in marine isoprene. Our first observations of isoprene in the winter are very important for characterizing the seasonal cycle of isoprene emissions based on observations. Our discussion of isoprene is less than that of DMS, because we did not measure isoprene precursors and related substances. We made the following changes to the manuscript in order to highlight the importance of isoprene surface measurements:

Page 21, line 493: "The results of measurements in the surface ocean during the stormy and mostly dark winter season in the Southern Ocean will be valuable for future atmospheric aerosol chemistry model studies, as they will not need to rely any longer on pure assumptions."

We think the importance of isoprene is highlighted in the conclusions already. Nonetheless, we started a new paragraph in line 547 to address isoprene only and separating it from DMS, in order to emphasize the importance of isoprene.

7. I am curious about the amount of trace gas emission during the winter season of the Southern Ocean. Is it possible to estimate it. And how?

Yes, it can be estimated by multiplying the calculated trace gas fluxes in the manuscript by the number of days in winter and the sea ice-free area of the Southern Ocean, assuming a uniform set of concentrations and wind speeds. These assumptions are of course not true, so the scaled-up estimate of winter season emissions is uncertain and should be used with caution. Nonetheless, we can get an idea of the scale. The scaled-up amount of DMS and isoprene emitted during winter in the Southern Ocean are calculated to be 0.013 Tg S and $0.39*10^{-3}$ Tg C, respectively. Compared with the annual mean emissions south of 60°S of  1 Tg S (Lana et al., 2011), winter emissions account for about 1.3% of the annual DMS flux in the Southern Ocean. As these were the first winter time measurements of isoprene in the Southern Ocean there is no published value directly to compare to. In comparison to global annual emissions of 0.21 Tg C, calculated by Booge et al (2016), winter time isoprene emissions account for ~0.2% of global annual emissions.

8.  During the winter season of the Southern Ocean, the oxidation of trace gases is known to be slow under the dark and cold atmosphere. It means that the particle formation from the trace gases is not that easy. Could the authors comment on the role of trace gases in influencing the climate during the winter season of the Southern Ocean?

The reviewer is correct that radical concentrations such as OH, $NO_3$, BrO may not be significant in the dark and cold winter due to weak solar radiation in the Southern Ocean. The oxidation of trace gases is slower than in warm conditions with sufficient light. However, even if suppressed, trace gas emissions from the Southern Ocean during winter time could potentially affect the climate. Because DMS and isoprene are readily emitted into the atmosphere, they may accumulate in the winter troposphere due to a longer lifetime. The existence of this build up and its potential effect through transport and when the sunlight turns on in spring is unknown. The relationship between oxidation products (e.g., aerosols and clouds) in all seasons must be established if we want to fully understand the natural atmospheric background. Therefore, despite low reactivity, observations of all components of the air-sea system are needed in the winter as well as the summer season.

**References**

Bonsang, B., Polle, C., and Lambert, G.: Evidence for marine production of isoprene, Geophysical Research Letters, 19, 1129-1132, https://doi.org/10.1029/92gl00083, 1992.

Booge, D., Marandino, C. A., Schlundt, C., Palmer, P. I., Schlundt, M., Atlas, E. L., Bracher, A., Saltzman, E. S., and Wallace, D. W. R.: Can simple models predict large-scale surface ocean isoprene concentrations?, Atmospheric Chemistry and Physics, 16, 15, https://doi.org/10.5194/acp-16-11807-2016, 2016.

Shaw, S. L., Gantt, B., & Meskhidze, N.: Production and emissions of marine isoprene and monoterpenes: a review. Advances in Meteorology, 408696, https://doi.org/10.1155/2010/408696, 2010.

---

## Author Comment (AC2)

We thank Reviewer #2 for the detailed comments. Attached are the responses to the specific comments. All critical points raised have been considered and addressed/discussed as detailed in this document (blue). Every indication of lines and/or pages refer to the originally uploaded manuscript.

1. The abstract should focus on the research results of this paper rather than its significance.

We agree and added more results from our study to the abstract (line 16):

"We found that the concentrations of DMS from the surface seawater and air in the investigated area were  $1.03 \pm 0.98$  nmol-1 and  $28.80 \pm 12.49$  pptv, respectively. The concentrations of isoprene in surface seawater were  $14.46 \pm 12.23$  pmol-1. DMS and isoprene fluxes were  $4.04 \pm 4.12 \mu$ mol m-2 d-1,  $80.55 \pm 78.57$  nmol m-2 d-1, respectively. These results are generally lower than the values presented or calculated in currently used climatologies and models."

 Page 1, line 30-32, 'Here we focus on two typical marine biogenic gases, i.e. dimethylsulphide (DMS) and isoprene...' The article discusses not only marine biogenic gases but also related sulphur compounds, I suggest to add a sentence of the relevant sulfides here.

We totally agree and added the related sulfur compounds to the sentence:

"Here we focus not only on two typical marine biogenic gases, i.e. dimethylsulphide (DMS) and isoprene, which have a significant influence on aerosols and climate in remote areas of the world (Carpenter et al., 2012; Lovelock et al., 1972), but also on two related sulphur compounds, i.e. dimethyl sulphoniopropionate (DMSP) and dimethylsulphoxide (DMSO).

3. Page 2, line 53, At the end of this paragraph, the article has been discussing the change data of radiation, and it is recommended to add a concluding sentence.

Thanks for this recommendation. We added following sentence to the end of the paragraph:

"These previous studies clearly show the importance of DMS emissions and related atmospheric oxidation products, and point to the importance of understanding how global DMS concentrations and subsequent emissions vary over the course of the year and over longer time periods."

4. Page 2, line 55, 'accounting for 50% of all BVOCs coming from terrestrial ecosystems.' Is the information unclear here? Is it 50% of the species or the quantity?

Thank you for pointing out that this sentence is unclear. The 50% refer to the quantity of emissions. We changed the sentence accordingly, to be more clear:

"...accounting for 50% of all BVOC emissions coming from terrestrial ecosystems."

5. Page 2, line 58, 'Most isoprene in the atmosphere is produced by terrestrial ecosystems...' How much is the most here? It is recommended to add data. If not, please delete 'most'.

We have modified as follows:

"Most isoprene in the atmosphere is produced by terrestrial ecosystems (>99%, Guenther et al., 2006), but isoprene is also known to be produced in the ocean as well by different species of phytoplankton, seaweed (Shaw et al., 2010, Bonsang et al., 1992), and some species of marine bacteria (Exton et al., 2013)."

6. Page 2, line 61, Add a transition sentence between the source and flux sentences.

We added following transition sentence:

"Since atmospheric isoprene in remote regions of the open ocean are directly related to surface seawater isoprene concentrations (Bonsang et al., 1992), biological marine isoprene production directly influences the magnitude of emissions to the atmosphere."

7. Page 3, line 74-75, 'Therefore, trace gas fluxes are computed using measured wind speed, measured atmospheric concentrations, and measured seawater concentrations.' It is repeated with the method below, it is recommended to delete it here.

The reviewer is right and the sentence was deleted.

8. Page 12, line 314, '...over the cruise track area are lower than the climatological concentrations,' How much percent lower is Lana's concentration?

We have calculated that our values are  $2.2\pm0.4$  times lower than the values of Lana et al. (2011) in the open ocean regions of our cruise track. However, in the coastal area the describes values in Lana et al. (2011) are  $1.6\pm1.0$  lower than our measured values in the same area.

9. In general, headings should not be followed directly by figures and tables.

Thanks for pointing that out. We have changed the location of figures and tables which directly follow the heading in the current version.

10. What does DOY stand for? Shouldn't it be DAY?

DOY is correct and it means "day of year". Unfortunately, we did not introduce it in this manuscript. We have added the meaning of DOY at its first appearance in the method section.

"...on 18 July 2019 (199th day of year, DOY 199),..."

**References**

Bonsang, B., Polle, C., and Lambert, G.: Evidence for marine production of isoprene, Geophysical Research Letters, 19, 1129-1132, https://doi.org/10.1029/92gl00083, 1992.

Carpenter, L. J., Archer, S. D., and Beale, R.: Ocean-atmosphere trace gas exchange, Chemical Society Reviews, 41, 6473-6506, https://doi.org/10.1039/C2CS35121H, 2012.

Exton, D. A., Suggett, D. J., McGenity, T. J., and Steinke, M.: Chlorophyll-normalized isoprene production in laboratory cultures of marine microalgae and implications for global models, Limnology and Oceanography, 58, 1301-1311, https://doi.org/10.4319/lo.2013.58.4.1301, 2013.

Lana, A., Bell, T. G., Simo, R., Vallina, S. M., Ballabrera-Poy, J., Kettle, A. J., Dachs, J., Bopp, L., Saltzman, E. S., Stefels, J., Johnson, J. E., and Liss, P. S.: An updated climatology of surface dimethlysulfide concentrations and emission fluxes in the global ocean, Global Biogeochemical Cycles, 25, https://doi.org/10.1029/2010gb003850, 2011.

Lovelock, J. E., Maggs, R. J., and Rasmusse.Ra: Atmospheric dimethyl sulfide and natural sulfur cycle, nature, 237, 452-&, https://doi.org/10.1038/237452a0, 1972.

Shaw, S. L., Gantt, B., & Meskhidze, N.: Production and emissions of marine isoprene and monoterpenes: a review. Advances in Meteorology, 408696, https://doi.org/10.1155/2010/408696, 2010.